# Barriers and Motivators toward Childhood COVID-19 Vaccination: A Cross-Sectional Study Conducted among Saudi Population

**DOI:** 10.3390/medicina59122050

**Published:** 2023-11-21

**Authors:** Hind M. AlOsaimi, Ali M. Alqahtani, Nadia M. Alanazi, Nouf N. Alotibi, Mohammed S. Alrazog, Hanoof A. Aljameel, Raghad M. Alshehri, Sarah J. Alhusayni, Mohammed K. Alshammari

**Affiliations:** 1Pharmacy Services Administration, King Fahad Medical City, Riyadh Second Health Cluster, Riyadh 12211, Saudi Arabia; amalqahtani1@kfmc.med.sa; 2Department of Pharmacy Rumah General Hospital, Riyadh 12211, Saudi Arabia; naianazi@moh.gov.sa; 3Department of Pharmacy, Qassim University, Qassim Region, Buraidah 56218, Saudi Arabia; noufottibi@gmail.com (N.N.A.); hanoofaljameel@hotmail.com (H.A.A.); 4Department of Pharmacy, Prince Sultan Military Medical City, Riyadh 12211, Saudi Arabia; malrazog@psmmc.med.sa; 5Department of Pharmacy, King Khalid University, Abha 62217, Saudi Arabia; raghadalshehri.98@gmail.com; 6Department of Pharmacy, Taibah University, Medina 41311, Saudi Arabia; sarah-alhussaini@hotmail.com

**Keywords:** COVID-19, vaccination, children, parents, hesitancy, Saudi Arabia

## Abstract

*Background and Objectives:* In 2020, one of the most important steps that were made was to give priority to the development of a COVID-19 vaccine to prevent the rising incidence of COVID-19 from continuing to rise. However, globally, there is a variable acceptance of the level of the COVID-19 vaccine. This study aims to explore Saudi parents’ willingness to vaccinate their children against COVID-19. *Materials and Methods*: This was a cross-sectional study; the online questionnaire was used to investigate the views of Saudi parents with children aged under 18 toward the immunization of their children against COVID-19. The data were gathered from 10 May 2022 to 31 October 2022. The data analysis uses SPSS version 20. A *p*-value of 0.05 or lower was regarded as statistically significant. *Results:* A total of 978 Saudi parents participated in this study. Most of the respondents were from the age group of 36–45 years with the educational qualification of high school and bachelor’s degree. Overall, it was observed that the majority, 98.2% of the respondents, disclosed that they needed more information (*p* = 0.004) about COVID-19 vaccine safety among children so that they could decide whether to vaccinate their child. About 91.4% of parents mention that vaccination against COVID-19 is not necessary for children (*p* = 0.001). About 68.3% of respondents agreed that getting vaccinated against COVID-19 could help Saudi Arabia control COVID-19 (*p* = 0.007, RI = 0.76). In terms of negative attitudes toward COVID-19 vaccination, 71.3% think that there will be severe side effects (*p* = 0.019, RI = 0.75); 67.7% think that the vaccine’s protection will only last for a short time (*p* = 0.055, RI = 0.72); 80.1% said they were afraid of getting vaccinated because of needle fear (*p* = 0.045, RI = 0.76), and 41.2% said lack of time was the main barrier to not vaccinating their child. *Conclusions:* Parents expressed concerns regarding the safety and efficacy of the COVID-19 vaccine, which might be some of the main factors influencing their decision to vaccinate their children. It is the need of the hour to take action to communicate, educate, and intervene with Saudi parents to enhance COVID-19 vaccination compliance rates across the board.

## 1. Introduction

The first case of the COVID-19 outbreak was reported on 31 December 2019 in Wuhan, China, and on 13 January 2020, Thailand reported its first case of COVID-19 [1]. Since then, COVID-19 has become a global pandemic [2]. Saudi Arabia is the second most populated country in the Gulf region, with a population of more than 34 million, of whom about 37% are expatriates [3]. The first case of COVID-19 was reported on 2 March 2020; before that, Saudi Arabia was one of the first countries to take preventative measures to stop the propagation of COVID-19 in the Kingdom [4]. Travel restrictions, strict immigration checks, movement control, implementation of sanitization and hygiene practices at workplaces (i.e., use of sanitizer and masks), suspending academic and religious activities, and reducing the number of workers in workplaces were some of the main preventive actions implemented [5]. Perhaps, due to these timely measures, the prevalence of COVID-19 was observed to be lower compared to the neighboring Gulf countries [6].

In 2020, The World Health Organization (WHO) declared COVID-19 a global pandemic, and timely development of a COVID-19 vaccine was prioritized to prevent the rising incidence of COVID-19 [7]. Regardless of these global efforts for vaccine development, however, there was a considerable number of individuals worldwide who refused COVID-19 vaccination [8]. According to the Strategic Advisory Group of Experts on Immunization (SAGE) of the World Health Organization (WHO), vaccine hesitancy (VH) is defined as a “delay in acceptance or refusal of vaccination despite the availability of vaccination services” [9]. WHO emphasized the necessity of the national and international stakeholders to conduct campaigns on the significance of COVID-19 immunizations so that vaccine hesitancy among the public could be reduced and willingness to vaccinate themselves and their children could be increased [10].

High uptake rates are necessary for a successful immunization response [11]. Both the acceptance of vaccines and the demand for them are dependent on the community’s perceptions toward vaccination [12]. Recent research has shown that equal access to vaccination across all demographic groups is difficult to achieve because of the complexity of human behavior and the fact that it varies over geography and time. Therefore, the groups willing to get vaccinated should be prioritized for vaccination, and side-by-side efforts should be made to educate and create awareness among those who may have negative perceptions toward vaccination [13]. The COVID-19 vaccine developed by Pfizer and BioNTech (AstraZeneca^®^) can now be administered in Saudi Arabia specifically to children aged 5 to 11 after receiving approval from the Saudi Food and Drug Authority. However, the willingness of parents toward vaccination is very important; it is observed that overall, in the last few decades, there has been an increase in parents’ refusal of immunizations for their children, and this has led to an increase in the number of vaccine-preventable diseases [14]. Belief in conspiracy theories (such as the idea that vaccines might harm or sterilize children or contain contagion) are some of the main factors strengthening vaccine refusal [15,16]. Since the beginning of its development, the COVID-19 vaccination has been linked to many conspiracies as well as rumors concerning its safety and effectiveness [17].

In an attempt to regulate the situation in Saudi Arabia, about 30 million doses of the COVID-19 vaccine (AstraZeneca) have been distributed [18]. It is evident from recent studies that parents have a relatively positive attitude toward vaccinating their children against COVID-19, with an overall vaccine hesitancy ranging between 33% [19] and 21% [20]. To overcome this information gap and ensure that successful vaccine interventions are properly designed and fine-tuned for the community, it is imperative that such attitudes and intentions be understood. This understanding is a pressing necessity [21]. The Health Belief Model may provide the stakeholders with a suitable set of measures for effective interventions to modify parents’ behaviors and negative perceptions toward COVID-19 vaccination [22]. If there is a widespread hesitation to get vaccinated against COVID-19, it is critical to identify the factors (i.e., attitudes or reasons) that lie behind the resistance toward vaccination. Public education programs (such as “do it for the herd”) can be helpful to some extent; however, identifying factors leading to vaccination hesitancy is essential to improve overall vaccine uptake [23]. The aim of this study is to identify the barriers to COVID-19 vaccination among Saudi parents so that timely interventional measures can be planned to attain “Herd Immunity” in the Saudi Region.

## 2. Materials and Methods

This cross-sectional study was conducted using an online questionnaire to investigate the Saudi parents’ perception of COVID-19 immunization for their children under 18. Online data collection is proven to be an efficient, timely, and rapid method that enables researchers to collect insight on the topic under consideration in a short time with a higher response rate [24,25].

### 2.1. Tool Development and Validity

A structured questionnaire was developed [26,27,28,29], which was subjected to a validation process explained in detail below [30,31,32]:Step one: Preparation of draft and any permission required;Step two: Content validity;Step three: Finalization of recommendations;Step four: Face validity;Step five: Final version of the tool.

During the first step, a draft of the study tools was finalized, and permissions of use were taken under the condition that the relevant citations would be given when this paper was published. Wild et al. (2005) stated that, ideally, the authors of the original tool should be invited to participate in the validation of the tool [30]. The entire pattern and the contents of each section were discussed in detail and were well understood by the lead researcher and colleagues. To avoid ambiguities, the draft version of the study tool was submitted to five experts from clinical care and academic settings to finalize the contents of the study tool. Some changes were suggested by these members, which aimed to simplify the language or presentation of the questions and the selection of the option to respond to the study questions. The finalized version was forwarded to the members again, and all agreed with the final version of the tool and recommended this tool be used for an online survey.

The final version was proofread, and some typing errors were then rectified and submitted to face validity of 30 respondents. Internal consistency was measured with coefficient alpha or Cronbach’s coefficient alpha (CA) [33,34]. In addition, CA is easy to interpret as its value ranges from 0 to 1, and the CA value for this tool was 0.79. Furthermore, to address any further concerns about the tool’s content, its adequacy was measured using Bartlett’s test of sphericity. The Kaiser–Meyer–Olkin measure of sampling adequacy is an effective technique for judging content adequacy. If the Kaiser–Meyer–Olkin value is higher than 0.6, it demonstrates that the contents of the instrument are satisfactory to meet the study needs [35,36].

### 2.2. Construct of Study Tool

An overview of the objectives, methods, confidentiality, and anonymity agreements was provided on the questionnaire’s cover page. The questionnaire contained three sections. The first section targets demographic characteristics, including age, gender, marital status, educational level, type of work, history of seasonal influenza vaccination, family history of COVID-19, the informational sources regarding COVID and its vaccine, and child’s age. Questions in the second section are intended to gauge parents’ perceptions of the decision not to vaccinate their kids against COVID-19. Questions in the third section gauge parents’ attitudes toward the COVID-19 vaccination.

### 2.3. Data Collection Procedure

The online forms were circulated over various social media websites. Online data collection is an effective method to collect data and plan the required interventions, and various international studies conducted on vaccine hesitancy for COVID-19 vaccination have utilized online data collection procedures using different methods [37,38,39,40]. In addition, a QR code was printed on a banner explaining participation in this online survey and pasted with permission in the ER departments of the hospitals and nearby pharmacies. QR code is a very convenient method of participation in a study, and it will take the respondent to the questionnaire page with one click. In addition, the link to the online data collection tool was also shared in different social media groups to gather data. Additionally, respondents were requested to share the form with other people in their immediate circle to share their opinions about the questions regarding COVID-19 vaccination among children. The data were gathered over a period of six months, beginning on 10 May 2022 and ending on 31 October 2022.

### 2.4. Inclusion and Exclusion Criteria

All Saudi residents who were married and had at least one child under 18 were included in this study. All non-Saudi parents are among this study’s exclusion criteria. Unmarried individuals and parents without children were excluded from this study.

### 2.5. Sample Size

Using Rao soft^®^, the sample size was calculated to be 377 as the minimal effective sample with a 95% confidence interval, 5% margin of error, and a 50% response distribution. An online questionnaire was used to approach potential participants. The distribution of the form was performed via social media platforms or community online social groups.

### 2.6. Statistical Analysis

This data analysis uses SPSS version 20. Both descriptive (frequency, mean, standard deviation) and inferential statistical techniques (chi-square test) were used to analyze the data. To determine the relationship between the groups and variables, the chi-square test was used. Furthermore, the relative index was also estimated to identify the top priorities of the parents for their willingness to vaccinate their children. A *p*-value of 0.05 or lower was regarded as statistically significant.

### 2.7. Ethical Approval

The ethical approval for this study was granted by the institutional review board (IRB) at King Fahad Medical City (IRB Log Number: 23-220, Dated 29 May 2023). A brief introduction to this study was given on the title page of the questionnaire. An online consent was taken before the response was collected from the respondent. It was entirely voluntary to participate. Since no information was gathered that could reveal the respondents’ identities, the confidentiality of the responses was guaranteed.

## 3. Results

A total of N = 978 Saudi parents opted to respond using an online survey form and share their views toward vaccinating their children against COVID-19. The demographic information about the participants is shown in Table 1. The majority (45.6%) of the respondents were from the age group from 36 to 45 years (45.6%), with relatively more men (50.1%) than women. All the respondents who participated in this study were literate, and most disclosed having high school and bachelor’s education. In terms of occupation, most had their own businesses (543 [55.5%]). In response to a very specific question regarding flu shots, about 49.3% disclosed noncompliance with the routine seasonal flu vaccination, while 52.6% disclosed that they or someone in their family suffered from COVID-19. Details are described in Table 1. The prime source of information for the majority (73%) of the parents for COVID-19 immunization was social media sites like Facebook, LinkedIn, etc., followed by friend and family groups (70.6%). In addition, 55.7% of parents learned about COVID-19 immunization from television and other electronic media, whereas 59% of parents received messages about COVID-19 vaccination from health authorities. Only 35.8% of parents acquired information from print media, such as newspapers and magazines.

The next section of this study’s questionnaire aimed to explore the parents’ willingness to vaccinate their children against COVID-19 (Table 2). In response to the question of whether the COVID-19 vaccine was safe and risk-free for children, 98.2% of parents answered that they needed more information about COVID-19 vaccine safety so that they could decide effectively to vaccinate their children with COVID-19 vaccine (*p* = 0.004). In addition, further information that parents needed to decide on COVID-19 vaccination for their children was proof of the COVID-19 vaccine’s advantages (95.1%, *p* = 0.015). Otherwise, most of the parents, 91.4%, had reservations against COVID-19 vaccination, assuming that it was not necessary for children (*p* = 0.001). Details are shown in Table 2.

Therefore, the sum of responses for this specific question may not be 100%.

This section was the main part of this study, aiming to explore the potential barriers and motivators to parents’ willingness to vaccinate their children against COVID-19. Analysis has revealed that 83.3% of the parents were aware of the fact that the National Health Services Department of Saudi Arabia has ensured the availability of the COVID-19 vaccine in the Kingdom (RI = 0.80; Rank 2; *p*-value < 0.001) and it is convenient for them to vaccinate their child. However, they emphasize that their willingness to vaccinate their children is the main factor in case they want to vaccinate their children against COVID-19 (RI = 0.84; Rank 1; *p*-value < 0.001). Regardless of this fact, the majority, 68.3%, were optimistic that timely vaccination against COVID-19 among children will assist in preventing COVID-19 in Saudi Arabia (RI = 0.76; Rank 3; *p*-value < 0.007). Almost 80.0% of the parents were observed to be reluctant toward vaccination due to the needle fear their children had (RI = 0.76; Rank 3; *p*-value < 0.045), and 71.3% disclosed fear of side effects (RI = 0.75; Rank 4; *p*-value < 0.019) as the main factor hindering COVID-19 vaccination. Other concerns that were limiting willingness toward COVID-19 vaccination were parents’ perception that the effect of COVID-19 vaccine might last for a short time in children (RI = 0.72; Rank 6; *p*-value < 0.055) and lack of time to bring children for vaccination (RI = 0.64; Rank 7; *p*-value < 0.029). Nearly 74.8% disclosed that they had sufficient family support in case they wanted to get their child vaccinated. Details regarding the barriers and motivators for vaccination are shown in Table 3.

## 4. Discussion

This is perhaps the first Saudi study that has reported vaccine hesitancy among Saudi parents toward COVID-19 vaccination and their willingness to vaccinate their children. One of the main strengths of this study is the large sample size of 978 parents, which has enabled the researcher to explore the severity of the COVID-19 vaccine in an effective manner. The findings of this study revealed that parents were found relying on social media as a main source of information regarding COVID-19 and its vaccination. Overall, the majority of the parents insisted on the need to provide more information regarding the COVID-19 vaccine and its safety in children; this has resulted in reservations among more than 90% of the parents to vaccinate their children against COVID-19. In addition, needle fear and the lack of benefits of COVID-19 vaccination among children were some of the main barriers revealed by this study.

For the effective outcome of any vaccine program, it is essential that all the members of the community receive the required doses so that the required HERD immunity can be attained. In addition, to achieve this vital task, it is important to assess willingness for vaccination in the community [41]. The current study is a similar effort aiming to explore Saudi parents’ willingness to vaccinate their children aged 18 years and younger against COVID-19. Unfortunately, the results of this study have shown a significant level of reluctance/hesitancy among Saudi parents to vaccinate their children aged 18 or younger against COVID-19. Parents who participated in this study were found to emphasize their willingness on whether they wanted to vaccinate their child or not. In addition, the majority reported that if they wanted to go ahead with COVID-19 vaccination for their child, they would have sufficient time and family support for this decision.

The vast majority of parents have reservations about vaccinating their offspring. It could be possible that the skepticism of parents regarding processes of vaccine development and approval highlights the need for ongoing transparency and active public education regarding the development and approval process carried out by the United States Food and Drug Administration, the Advisory Committee on Immunization Practices, and the Centers for Disease Control and Prevention (CDC) [42]. It is crucial to underline the safety profile of vaccines for children, which can be derived from both vaccination trials and post-approval data that will definitely assist in achieving the required vaccination rates for the children [43]. In this situation, there is an urgent need to design interventions that are tailored to Saudi parents to address specific concerns on safety, efficacy, development, and after-vaccination benefits so that national indicators for vaccination can be achieved [44]. Specifically, the level of refusal for COVID-19 vaccination among Saudi parents is somewhat lesser in proportion to the studies that have addressed similar research in France, Germany, Italy, Portugal, the Netherlands, and the United States [45,46,47] and comparable to reports from Denmark and Australia [48].

One of the main concerns that is observed in this study is perhaps the lack of will to vaccinate the children. About 83.3% of the parents shared that they were aware that the vaccine for COVID-19 was sufficiently available in the Kingdom; however, if they were willing to vaccinate their child, then they would definitely consider visiting the health center for COVID-19 vaccination (RI = 0.84; Rank 1; *p*-value < 0.001). Parents’ willingness to vaccinate their children was observed to be the main barrier to COVID-19 vaccination. The results from this study reported a unique parents’ perspective, “If I am willing, I will vaccinate my child”. Almost 80.0% of the parents were observed to be reluctant about vaccination due to the needle fear in their children (RI = 0.76; Rank 3; *p*-value < 0.045), and 71.3% disclosed fear of side effects (RI = 0.75; Rank 4; *p*-value < 0.019) as the main factor hindering COVID-19 vaccination. The majority of people who refused the COVID-19 vaccine cited concerns over vaccine safety, effectiveness, and the benefits of vaccinating children as their justifications [49]. The parents’ concerns regarding the safety and risk of any unproven harm to their children are perhaps the main factors that will hinder their willingness to COVID-19 vaccination. Almost similar initial concerns were reported in other parts of the world; however, upon observing the severity of infection among the geriatric population, European parents feared the worst consequences of COVID-19 for children, and perhaps the seriousness of the disease has elevated their trust in the effect of the vaccine, and they were more receptive to COVID-19 vaccine for children [50]. Large-scale public education efforts have been launched to encourage people to get vaccinated; it is possible that these campaigns have influenced people’s faith in vaccines [20,51], and similar approaches can be adopted by Saudi health authorities to increase COVID-19 vaccination among children.

In addition, for a newly developed vaccine for COVID-19, similar results of hesitancy were shown by the parents from the US who requested more reliable information, which will definitely increase their confidence in COVID-19 vaccination [38,52,53]. While some participants thought children were less vulnerable to COVID-19, others pointed out that youngsters were more likely to experience long-term negative effects. Parents said that additional proof is needed to validate the vaccine’s safety for young patients. Moreover, the lack of multinational clinical trials and the efficacy of the mRNA vaccine [54] are reported to be the other factors hindering COVID-19 vaccination among children. In addition, the majority of the parents disclosed that they were relying on social media as a source of information for COVID-19 vaccination; in this, the COVID-19 vaccine was massively criticized with false accusations. It is essential to investigate how social media influences the perspectives and worries of parents, as well as what they may believe about a new pandemic or health concern in general, in order to determine the various factors that may affect parents’ levels of tolerance to new COVID-19 vaccination [55,56]. The elements that influence the willingness of parents to vaccinate their children.

These findings provide information that can serve as a baseline for future vaccination programs, which can help boost immunization rates. In order for public health officials to establish successful immunization strategies to improve vaccine uptake for the purpose of preventing and controlling COVID-19, it will be helpful for them to investigate both the obstacles to vaccination and the factors that facilitate its use [57]. The fact that the participants are active on many anti-vaccination Facebook sites suggests that the participation of individuals in anti-vaccination is more than just a consequence of Facebook’s recommended algorithm [58]. Previous studies in the literature that did not utilize social media to investigate reasons for reluctance were consistent with the reasons for hesitancy, which suggests a consistency in hesitancy reasons regardless of whether they are expressed on social media or not [59]. According to the findings of this study, users of social media in different parts of the world exhibited varying levels of good and negative emotions, and these feelings changed over the course of their interactions with the platform. According to the findings of one study, public opinion is dependent not only on the government but also on the company that makes the vaccine [60].

### Limitations of This Study

This study has utilized the online data collection method, which is utilized by most of the international studies exploring vaccine hesitancy toward COVID-19 vaccination for urban population [37,38,39,40]. In addition, the prime focus of this research was on the Saudi population residing in the urban areas of Saudi Arabia; therefore, it is possible that the level of hesitancy toward COVID-19 vaccination may be higher or lower in comparison to the perspective of the study population reported in this study. Regardless of this limitation, this study manages to represent the barriers and facilitators toward COVID-19 vaccination in a very effective manner and will assist the public health stakeholders in planning respective interventions effectively.

## 5. Conclusions

For health policy-makers and healthcare professionals organizing vaccination programs, this study gives current information on parents’ opinions about immunizing their children. Concerns expressed by respondents about a prospective COVID-19 vaccine serve as crucial targets for potential international educational campaigns, which are the need of the hour to increase rates of vaccination. Saudi parents’ main excuse not to vaccinate their children with the COVID-19 vaccine was their self-perceived willingness, followed by the concerns about the safety and potential adverse effects and the perception that their child was not at risk for COVID-19 were the subjects of the respondents’ greatest concerns. The parents were heavily influenced by social media. There is an urgent need to educate patients with evidence-based information so that the effect of unreliable information that is available on various social media platforms can be neutralized. Also, as a scientific community, we must take action to communicate, educate, and intervene to enhance COVID-19 vaccination compliance rates across the board. Overall, most parents reported that if they had time to go for COVID-19 vaccination for their children and if they were willing to go for this decision, there would be sufficient family support.

### Recommendations

Tourism (Hajj and Umrah) is one of the main sources of revenue generation for Saudi Arabia. In this case, the possibility of any outbreak will affect not only the Saudi economy but also the health of the residents. Moreover, in Saudi Arabia, more than 85.0% of the population resides in rural areas; thus, there is an immediate need for tailored communication campaigns or approaches to assist in improving health literacy among the residents to enhance acceptance of the COVID-19 vaccination.

In addition, there is an immediate need to train healthcare providers as well so that they can act as effective educators and counselors whenever they identify parents or adults who are hesitant toward COVID-19 vaccination. Moreover, having a trained health task force will also assist in neutralizing the disinformation/negative claims that are shared on various social media sites. Vaccine hesitancy has emerged as a global problem. Since the conditions of every person and society are different, it is essential for Saudi stakeholders to identify and measure vaccine acceptance or reluctance as well as access constraints. In order to combat misinformation, effective community participation and communication tactics are required, with coercive measures being employed as a last choice following less restrictive and trust-building initiatives.

## Figures and Tables

**Table 1 medicina-59-02050-t001:** Socio-demographics of participants.

Sociodemographic Characteristics of the Parents (n = 978)	N	%
Age group		
<25 years	35	3.6
25–35 years	250	25.6
36–45 years	446	45.6
>46 years	247	25.3
Gender		
Male	490	50.1
Female	488	49.9
Marital status		
Married, having at least one child	937	95.8
Divorced, having children	41	4.2
Educational level		
None	0	0.0
High school or below	361	36.9
Diploma	207	21.2
Bachelor’s	340	34.8
Master’s	60	6.1
PhD	10	1.0
Type of work		
Private Sector	120	12.3
Public Sector	217	22.2
Non-profit Organization	543	55.5
Retired	97	9.9
History of seasonal influenza vaccination		
No	482	49.3
Yes, within one year	344	35.2
Yes, beyond one year ago	153	15.6
Family history of COVID-19		
Yes	514	52.6
No	464	47.4
Child age		
3 years or under	206	21.1
4–7 years	200	20.4
8–12 years	394	40.3
<18 years	179	18.3
From where you are accruing information about COVID-19 and its vaccination *		
Facebook, social media, LinkedIn, etc.	713	72.9
Newspapers, magazines, print media	350	35.7
Television, electronic media	545	55.7
Friend and family/friend groups	690	70.6
Messages from health authorities	577	59.0

* Multiple options were given to the respondents to disclose their responses.

**Table 2 medicina-59-02050-t002:** Factors influencing willingness.

What Do You Think About Why Parents May Not Consider Vaccinating Their Child Against COVID-19	Yes	No	*p*-Value
Parents may perceive children are not at risk of COVID-19 and, therefore, do not need vaccination	894	84	<0.001
Parents need more information before I decide about COVID-19 vaccine for my children	940	38	0.005
Parents need more evidence for COVID-19 Vaccine and its benefits for children	930	48	0.015
Parents need more information about the safety and risk of COVID-19 vaccine in children	960	18	0.004

Chi-square. *p*-value < 0.005 was considered statistically significant.

**Table 3 medicina-59-02050-t003:** Parents’ perception about their children’s vaccination.

Statement	Strongly Agree	Agree	Disagree	Strongly Disagree	RI	Ranking	*p*-Value
COVID-19 vaccination is effective in protecting your child from COVID-19	250	200	373	155	0.63	8	0.079
Timely vaccination of COVID-19 among children will assist preventing COVID-19 in Saudi Arabia	455	213	193	117	0.76	3	0.007
There is sufficient supply of COVID-19 vaccine and booster dose in Saudi Arabia	508	273	100	97	0.8	2	<0.001
Fear of side effects hinders my decision to vaccinate my child	413	284	155	126	0.75	4	0.019
The protection of COVID-19 vaccines will last for a short time in children	412	250	113	203	0.72	6	0.055
Due to needle fear, I am reluctant to vaccinate my child against COVID-19	303	480	120	75	0.76	3	0.045
You do not have time to take your child for COVID-19 vaccination	213	190	500	75	0.64	7	0.029
Family members would support you in having your child take up COVID-19 vaccination	342	390	100	146	0.74	5	0.029
Having the child receive COVID-19 vaccination is easy for you if you want them to	597	218	70	93	0.84	1	<0.001

No response was recorded for neutral option; relative index ranking was performed, and level of significance was tested using chi-square. *p*-value < 0.005 was considered statistically significant.

## Data Availability

Available on request.

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
