# Peer review of "Barriers and Motivators toward Childhood COVID-19 Vaccination: A Cross-Sectional Study Conducted among Saudi Population"

_medicina, 2023, doi:10.3390/medicina59122050_

Round 1

Reviewer 1 Report

Comments and Suggestions for Authors

A brief summary

The article is interesting, potentially good for publishing but some parts are a bit confusing and some are even unnecessary. It needs to be read carefully and revised by authors due to typos.

General comments:

The article needs a thorough proofreading of the English language by a professional. There is no need for double full stops in the end of the sentence (before and after the reference). The introductory part is too long and could be shortened by providing only essential information related to the work. Authors are advised to pay attention to what they use the term COVID-19, covid and SARS-COV 2. It is suggested for authors to put subtitles in Materials and Methods without colon (:).Materials and methods are presented in detail, which is very good for further research, although some parts could be shortened a bit for example Ethical approval. It is also necessary to decide how many decimal places in the results will be expressed (eg Table 3 - either 0.8 or 0.80 if two decimal places are given everywhere). The beginning of the Discussion is a bit confusing and needs to be redone as well as a Conclusion section.

Specific comments:

Line 24  Instead of covid vaccination put COVID-19 vaccination

Line 25-27            Please rewrite this  two sentences due to better understanding

Line 27-29            The sentence for ethical approval does not belong in the abstract

Line 30                  Please exclude „N=“ from this sentence. It is enough to say for example „There were 978 Saudi parents ...“

Line 35                  Please put p=0.001 instead of p 0.001

Line 56-60            Perhaps it would be better to transform this sentence so that there are no brackets

Line 60-63            Combine this two sentences into one

Line 67                  Maybe „however“ is a better word here than „unfortunately“

Line 89                  Please rephrase this sentence without using slash (/) or brackets

Line 132-136       Maybe it would be better to retell this part in sentences, but if the authors think that it is clearer this way, it can stay that way

Line 195-196       Please combine this two sentences

Line 208                Please address to Instructions for authors for preparing (modification) of Tables (or use template that is intended for this journal)

Line 210                Children are listed in this table with a capital letter which is unnecessary

Line 214                disagree needs one more quotation mark

Line 232                4.0 Discussion does not need a colon (:), and it could only be stated as a 4. (not 4.0) (this also refers for the Conclusion)

Line 249, 282      Those Tables are the Results and not the Discussion

Line 233-238       This section needs to be rewrited to make it more clear. It is a bit confusing.

Line 238                „..social environment in which person is behaving“ – this expression is very strange, so maybe it would be good to state it differently

Line 354                In the conclusion, include a little more detailed facts obtained in your study.

Comments on the Quality of English Language

The article needs a thorough proofreading of the English language by a professional. 

Author Response

Dear Sir

Greetings

We are thankful to the respected reviewers for their expert comments on this paper.

Below is the response to the individual comments

Reviewer 1

General comments:

The article needs a thorough proofreading of the English language by a professional.

Reply:  Corrections done through out the paper

There is no need for double full stops in the end of the sentence (before and after the reference).

Reply:  corrected

The introductory part is too long and could be shortened by providing only essential information related to the work. 

Reply: Reduced up to 20%

Authors are advised to pay attention to what they use the term COVID-19, covid and SARS-COV 2.

Reply:  corrected

It is suggested for authors to put subtitles in Materials and Methods without colon (:).Materials and methods are presented in detail, which is very good for further research, although some parts could be shortened a bit for example Ethical approval.

Reply: Modified

It is also necessary to decide how many decimal places in the results will be expressed (eg Table 3 - either 0.8 or 0.80 if two decimal places are given everywhere).

Reply: Modified and corrected

The beginning of the Discussion is a bit confusing and needs to be redone as well as a Conclusion section.

Reply: revised

  Specific comments:

 Line 24  Instead of covid vaccination put COVID-19 vaccination

Reply:  corrected

Line 25-27            Please rewrite this  two sentences due to better understanding

Reply:  revised

Line 27-29            The sentence for ethical approval does not belong in the abstract

Reply:  removed

Line 30                  Please exclude „N=“ from this sentence. It is enough to say for example „There were 978 Saudi parents ...“

Reply:  corrected  

Line 35                  Please put p=0.001 instead of p 0.001

Reply:  corrected

Line 56-60            Perhaps it would be better to transform this sentence so that there are no brackets

Reply:  corrected

Line 60-63            Combine this two sentences into one

Reply:  combined

Line 67                  Maybe „however“ is a better word here than „unfortunately“

Reply:  corrected

Line 89                  Please rephrase this sentence without using slash (/) or brackets

Reply:  corrected

Line 132-136       Maybe it would be better to retell this part in sentences, but if the authors think that it is clearer this way, it can stay that way

Reply:  corrected

Line 195-196       Please combine this two sentences

Reply:  combined

Line 208                Please address to Instructions for authors for preparing (modification) of Tables (or use template that is intended for this journal)

Line 210                Children are listed in this table with a capital letter which is unnecessary

Reply:  corrected

Line 232                4.0 Discussion does not need a colon (:), and it could only be stated as a 4. (not 4.0) (this also refers for the Conclusion)

Reply:  corrected

Line 249, 282      Those Tables are the Results and not the Discussion

Reply:  corrected

Line 233-238       This section needs to be rewrited to make it more clear. It is a bit confusing.

Reply:  corrected

Line 238                „..social environment in which person is behaving“ – this expression is very strange, so maybe it would be good to state it differently

Reply:  corrected

Line 354                In the conclusion, include a little more detailed facts obtained in your study.

Reply:  corrected

Reviewer 2 Report

Comments and Suggestions for Authors

Thank you for sharing your manuscript on childhood COVID-19 vaccination. The following comments may help to improve the article:

L97/98: Please state in your manuscript which COVID-19 vaccine you are referring to. Were the 30 million doses distributed or administered? 

L124-126: Is your method not subject to selection bias as you are limiting your study population to parents having access to the internet? 

L159: Please be more specific in your manuscript regarding the variable "family history of COVID-19". Are you aiming to collect the onset(s) of COVID-19 among all family members? If so, how was COVID-19 confirmed? Please be more specific regarding the term "history"; which time frame does this cover? 

L159-162: You provided at least some variables of section 1. Will you give more detail on variables included in sections 2-4 at this stage of your manuscript? 

L164: How did you get access to parents' e-mail, WhatsApp and Facebook accounts? Please explain in your manuscript. Consider rephrasing "were requested" as study participation must be voluntary and this kind of selection of further participants does lack randomness.  

L169-170: You excluded parents having children older than 18 years before. What is your justification of including them at the end? 

L170-171: Regarding "keeping their youngest child in mind", what is the age group you targeted four your research; please be more specific. If parents had several children below the age of 18 years, did you collect data about all children or only the youngest one? Please clarify in your manuscript.

L176: How did you select the 978 parents?

L187: So some kind of written consent?

L188: How did you define your study respondent?

Comments on the Quality of English Language

Please see above.

Author Response

Reviewer 2

L97/98: Please state in your manuscript which COVID-19 vaccine you are referring to. Were the 30 million doses distributed or administered? 

Reply: Vaccine name added , it was distributed and respective citation is also provided

L124-126: Is your method not subject to selection bias as you are limiting your study population to parents having access to the internet? 

Reply: In Saudi Arabia use of mobile and internet doubled after COVID and in this way it was convenient for us to collect data without any face to face interaction. Which may result in higher refusal rate and may require more resources to collect data.  so we believe getting information from this cohort will be a better data set for policy measure and intervention for COVID19 vaccination acceptability.

L159: Please be more specific in your manuscript regarding the variable "family history of COVID-19". Are you aiming to collect the onset(s) of COVID-19 among all family members? If so, how was COVID-19 confirmed? Please be more specific regarding the term "history"; which time frame does this cover? 

Reply: Getting COVID testing information for family members was not the part of data collection.  It was just to get insight how many of the family members were affected from COVID-19

L159-162: You provided at least some variables of section 1. Will you give more detail on variables included in sections 2-4 at this stage of your manuscript? 

Reply:  Construct of study tool is revised

L164: How did you get access to parents' e-mail, WhatsApp and Facebook accounts? Please explain in your manuscript. Consider rephrasing "were requested" as study participation must be voluntary and this kind of selection of further participants does lack randomness.  

Reply: QR Codes were printed on a banner explaining participation in this online survey and pasted with permission in the ER department of the hospital, Near by Pharmacies. In addition link of the online data collection tool was also shared in different social media groups to gather data.

L169-170: You excluded parents having children older than 18 years before. What is your justification of including them at the end? 

Reply: 18 years and above are considered Adults so parent consent for vaccination may not be required that why excluded

L170-171: Regarding "keeping their youngest child in mind", what is the age group you targeted four your research; please be more specific. If parents had several children below the age of 18 years, did you collect data about all children or only the youngest one? Please clarify in your manuscript.

Reply: Questions were general in nature and we inquired parents perception for willingness to vaccinate all children that are age less than 18 year

L176: How did you select the 978 parents?

Reply: It was an online questionnaire based study QR Codes were printed on a banner explaining participation in this online survey and pasted with permission in the ER department of the hospital, Near by Pharmacies. In addition link of the online data collection tool was also shared in different social media groups to gather data.  

Reply: its an online survey we distributed study Q using different online platform as described in the method section

Reply: added in data collection section

L187: So some kind of written consent?

Reply:  online consent as described in the method section

L188: How did you define your study respondent?

Reply: inclusion criteria is presented in the method section

Reviewer 3 Report

Comments and Suggestions for Authors

This is a well-written manuscript with adequate details. Personal beliefs play an essential role here besides education, career, and wealth. A large part of the population is moderate in these aspects, leading to more mixed responses.

It is a nice article to evaluate reasons for vaccine hesitancy in children.

Author Response

Reviewer 3

Comments and Suggestions for Authors

This is a well-written manuscript with adequate details. Personal beliefs play an essential role here besides education, career, and wealth. A large part of the population is moderate in these aspects, leading to more mixed responses.

It is a nice article to evaluate reasons for vaccine hesitancy in children.

Reply:  Thanks Prof for the recommendations, we are highly obliged

Round 2

Reviewer 2 Report

Comments and Suggestions for Authors

Thank you for addressing some of my comments.

Comments on the Quality of English Language

Please see above. 

Author Response

Response to reviewer 2 comments

1- The authors should provide a detailed methodology for the study. Authors can look a manuscript (https://www.mdpi.com/2076-393X/10/12/2093) for structuring methods.
The authors have suggested that online distribution of the questionnaire is a good way to collect the data, as many people in Saudi Arabia are using the internet. However, this cannot minimize the risks of bias in this study. Many people in Saudi Arabia, particularly those who reside in rural areas, do not use the internet frequently, or they don't have accounts on social media. Moreover, it has been observed that people who use social media have more vaccine hesitancy as they come into contact with misleading information about vaccines. I believe that this study missed parents who do not use social media, and that is a very important limitation of the study. I would suggest the authors consider this limitation in the limitation section and discuss its impact in the discussion. Moreover, authors can provide references to other studies that have used similar methods of data collection in order to support their design.

Reply: relevant citation and supporting information is added in the methodology and limitation part

2- Why are the authors discussing results from China in detail at line 443? I believe that there are plenty of margins to improve the discussion section.

Reply: section is modified to be a general discussion and relevant citation are added

3- The authors should add a limitation and strength section before conclusions. There are various limitations associated with this study.

Reply:limitation section is added

4- Please rewrite the discussion. First few sentences should provide information on major results and then move on the importance of these results. The authors are encouraged to make statements on future directions for healthcare professionals, policymakers, and researchers

Reply: introductory paragraph is added and separate section on recommendation is added after conclusion section
